# The Impact of Valvuloarterial Impedance on Left Ventricular Geometrical Change after Transcatheter Aortic Valve Replacement: A Comparison between Valvuloarterial Impedance and Mean Pressure Gradient

**DOI:** 10.3390/jcm9103143

**Published:** 2020-09-29

**Authors:** Satoshi Yamaguchi, Yuka Otaki, Balaji K. Tamarappoo, Tetsuya Ohira, Hiroki Ikenaga, Jun Yoshida, Tarun Chakravarty, John Friedman, Daniel Berman, Florian Rader, Robert J. Siegel, Raj Makkar, Takahiro Shiota

**Affiliations:** 1Smidt Heart Institute, Cedars-Sinai Medical Center, Los Angeles, CA 90048, USA; Yuka.Otaki@cshs.org (Y.O.); balaji.tamarappoo@cshs.org (B.K.T.); Hiroki.Ikenaga@cshs.org (H.I.); Jun.Yoshida@cshs.org (J.Y.); tarun.chakravarty@cshs.org (T.C.); John.Friedman@cshs.org (J.F.); Daniel.Berman@cshs.org (D.B.); florian.rader@cshs.org (F.R.); Robert.Siegel@cshs.org (R.J.S.); Raj.Makkar@cshs.org (R.M.); takahiro.shiota@cshs.org (T.S.); 2Department of Epidemiology, Fukushima Medical University, Fukushima 960-1247, Japan; teoohira@fmu.ac.jp

**Keywords:** aortic valve stenosis, left ventricular mass, left ventricular remodeling

## Abstract

Increase in left ventricular (LV) mass develops as a compensatory mechanism against pressure overload in aortic valve stenosis. However, long-standing LV geometrical changes are related to poor prognosis. The LV geometrical change occurs after transcatheter aortic valve replacement (TAVR). The present study aimed to investigate the relationship between improvement in valvuloarterial impedance (Zva) and change in LV mass index (LVMI) and the ratio of LVMI to LV end-diastolic volume index (LVMI/LVEDVI). We compared these relationships to that between Zva and mean pressure gradient (MPG). Baseline and follow-up transthoracic echocardiograms of 301 patients who underwent TAVR from November 2011 to December 2015 were reviewed. Spearman correlation coefficient (ρ) was used to compare ΔLVMI and ΔLVMI/LVEDVI with Zva or MPG. The correlation between ΔZva and ΔLVMI (ρ = 0.47, *p* < 0.001) was superior to that between ΔMPG and ΔLVMI (ρ = 0.15, *p* = 0.009) (*p* for comparison < 0.001). The correlation between ΔZva and ΔLVMI/LVEDVI was statistically significant (ρ = 0.54, *p* < 0.001); in contrast, that of ΔMPG and ΔLVMI/LVEDVI was not. The improvement in Zva after TAVR was more closely related to LVMI and LVMI/LVEDVI reduction than MPG reduction.

## 1. Introduction

Aortic valve stenosis (AS) imposes pressure overload on the left ventricle (LV) and leads to LV geometrical alterations, such as increased LV mass relative to the LV cavity [1,2]. These LV geometrical alterations are compensatory mechanisms against pressure overload to keep the LV wall stress close to normal [1,3,4]. However, these long-standing geometrical changes are related to impaired left ventricular diastolic function, resulting in heart failure [5,6].

Valvuloarterial impedance (Zva) has been proposed as a new index to represent LV global load and consists of both systemic and valvular loads [7]. Zva can stratify the risk of poor prognosis and serious symptoms, such as syncope, in patients with severe AS [7,8,9].

LV mass reduction, considered as a reverse-remodeling, occurs after transcatheter aortic valve replacement (TAVR) [10,11,12,13]. However, large individual differences have been observed in LV mass reduction after TAVR [10,11,12,13]. Such a wide variation in LV mass alterations after TAVR cannot be explained merely by the change in aortic valve mean pressure gradient (MPG) after TAVR. Thus, we hypothesize that LV mass reduction after TAVR correlates better with Zva than with MPG. This study aimed to investigate the relationship between improvement in Zva and change in LV mass index (LVMI) and the ratio of LVMI to LV end-diastolic volume index (LVMI/LVEDVI). We compared these relationships to that between Zva and MPG.

## 2. Methods

### 2.1. Participants

We conducted a single-center retrospective observational study. A total of 1335 patients who underwent TAVR at the Cedars-Sinai Medical Center from November 2011 to December 2015 were reviewed. Aspirin was initiated at the timing of TAVR and continued at least one year. Of those patients, 801 patients had undergone comprehensive transthoracic echocardiography (TTE) at baseline and follow-up (≥90 days from TAVR). From these patients, we excluded 500 participants who met the following exclusion criteria: valve in valve TAVR (*n* = 58); moderate and severe aortic regurgitation (*n* = 130); moderate and severe mitral regurgitation (*n* = 172); moderate and severe mitral stenosis (*n* = 23); moderate and severe paravalvular leakage (*n* = 25); concomitant coronary artery stenting with TAVR (*n* = 10); transapical approach (*n* = 8); old myocardial infarction (*n* = 66); diagnosed ischemic, dilated, or hypertrophic cardiomyopathy (*n* = 7); and procedure-related myocardial infarction (*n* = 1). This left 301 patients were eligible for our analysis (Figure 1).

The ethics committee at the Cedars-Sinai Medical Center approved this study and waived the requirement for informed consent owing to the observational nature of the study. For the purpose of confidentiality, all data were de-identified and analyzed anonymously.

### 2.2. Echocardiography and Hemodynamic Parameters

TTE was performed at baseline and follow-up 426 (327, 788) (median with interquartile range) days after TAVR with the iE33 system (Phillips Medical Systems, Andover, MA, USA) by experienced technicians. A cardiologist reviewed results and performed the off-line measurements [14]. The following echocardiography parameters were measured in parasternal long, short-axis, and four- and two-chamber views and indexed to body surface area: LV mass index (LVMI), LV end-diastolic volume index (LVEDVI), stroke volume index (SVI), and effective orifice area index. Severe and moderate patient–prosthesis mismatch was defined as effective orifice area index < 0.65 cm^2^/m^2^ and 0.65 ≤ effective orifice area index < 0.85 cm^2^/m^2^ [15]. LVMI/LVEDVI was computed as a marker for LV concentricity [16,17]. The LV ejection fraction was measured using the biplane discs method [18]. LV mass was calculated using a cube formula (LV mass = 0.8 (1.04 ((LVDd + IVSth + PWth)^3^ – (LVDd)^3^)) + 0.6, in which LV end-diastolic diameter (LVDd), interventricular septum thickness (IVSth), and posterior wall thickness in diastole (PWTh) [18]. Continuous-wave Doppler was used to estimate the transvalvular peak pressure gradient using the Bernoulli simplified equation [19]. MPG was analyzed by tracing the continuous wave Doppler across the aortic valve. The continuity equation calculated the effective orifice area. SV was the product of the LV outflow tract area and LV outflow tract velocity–time integral measured by pulse wave Doppler [19,20]. Peak early and late mitral inflow velocities were measured using pulse wave Doppler, whereas early and late mitral annular velocities were measured using tissue wave Doppler [21]. Zva was calculated as (MPG + systolic arterial pressure)/SVI [7]. Cuff auscultation was used to obtain blood pressure at the examination on the same day as the echocardiographic examination. Aortic regurgitation at baseline and paravalvular leakage after TAVR were graded on a three-point scale: none, mild, moderate, or severe [15,22].

### 2.3. Subgroup Analysis

Severe aortic valve stenosis with low stroke volume (<35 mL/m^2^) and low mean pressure gradient (<40 mmHg) is known as low-flow low-gradient AS (LFLG-AS). LFLG-AS has been reported to be associated with higher baseline Zva [23]. We identified 22 patients with low-flow low-gradient aortic valve stenosis defined as <35 mL/m^2^ and MPG<40mmHg, and performed subgroup analysis. We also examined the relationship between baseline LV geometry and baseline Zva or baseline MPG and that between the change of LV geometry and ΔZva or ΔMPG.

### 2.4. Computed Tomography Acqusition Protocols, Reconstruction, and Analysis

In 21 patients, cardiac computed tomography (CT) at baseline was performed using a dual-source CT scanner (Definition; Siemens Medical Solutions, Forchheim, Germany). These datasets were processed using commercially available software (Vital Images Version 6.7, Minnetonka, MN, USA).

### 2.5. Statistical Analysis

Continuous variables with normal distribution and skewed distribution are expressed as mean ± SD and median (25%, 75%), respectively. Categorical variables are expressed as numbers and percentages. A Wilcoxon rank-sum test was used for the comparison of Zva, hemodynamics, and TTE parameters between baseline and follow-up.

Spearman correlation coefficient (ρ) was used to compare LVMI and LVMI/LVEDVI with Zva or MPG at baseline and on change. We used the Hittner, May, and Silver’s test to compare the two overlapping correlations in the dependent group [24].

To identify the factors associated with ΔLVMI and ΔLVMI/LVEDVI, we employed univariate linear regression models to examine the following 15 factors: ΔZva, ΔMPG, ΔSVI, clinical characteristics including age, sex, body mass index, past medical history (diabetes mellitus and hypertension), medications (angiotensin-converting enzyme inhibitor and/or angiotensin receptor blocker, and beta-blocker), systolic and diastolic blood pressure at follow-up, baseline LV geometry (LVMI or LVMI/LVEDVI), mild paravalvular leakage, and mild patient–prosthesis mismatch [4,14,24,25,26]. We also examined those factors in a multivariate linear regression model. Correlation coefficients and 95% confidence intervals were provided in the univariate and multivariate linear regression models.

LVMI, LVEDVI, and SV values assessed using echocardiography were compared with those using cardiac CT to validate echo parameters, and Spearman correlation coefficient (ρ) was used for analysis.

### 2.6. Statistical Software

All statistical tests were two-sided, and a *p* value < 0.05 was considered statistically significant. Statistical analysis was performed using R 3.4.1 (The Foundation for Statistical Computing), EZR 1.3.6 (Jichi Medical University Saitama Medical Center) [27], and R package “cocor 1.1.3” [24].

## 3. Results

### 3.1. Baseline Characteristics

The mean age was 82 ± 8 years. There were 174 men and 127 women. The risk of mortality based on a predictive model proposed by the Society of Thoracic Surgeons was 4.5 (3.0, 6.1). Table 1 shows demographic and procedure data.

### 3.2. Zva and Hemodynamics

Overall Zva and LFLG-AS values were significantly reduced after TAVR (Overall Zva, baseline 4.91 ± 1.11 vs. follow-up 3.87 ± 0.81 mmHg/mL/m^2^, *p* < 0.001; LFLG-AS, baseline 5.61 ± 0.86 vs. follow-up 4.14 ± 0.98 mmHg/mL/m^2^, *p* < 0.001; Table 2).

### 3.3. Echocardiographic Data

In the overall population, LVMI reduction was observed after TAVR (baseline 99 (83, 118) vs. follow-up 86 (69, 103) g/m^2^, *p* < 0.001; Table 1). LVMI/LVEDVI decreased after TAVR (baseline 1.51 (1.28, 1.78) vs. follow-up 1.27 (1.05, 1.51), *p* < 0.001).

The same findings—LVMI and LMVI/LVEDVI decrease—were obtained in the subgroup analysis of LFLG-AS as obtained in the overall population (Table 2).

### 3.4. The Relationship between Zva or MPG and LV Geometry (LVMI and LVMI/LVEDVI) at Baseline

There were no significant relationships between baseline MPG and LVMI nor LVMI/LVEDVI (Table 3). There were no significant relationships between baseline Zva and baseline LVMI, but baseline LVMI/LVEDVI (ρ = 0.29, *p* = 0.001; Table 3).

In LFLG-AS, there was no significant relationship between baseline MPG and LVMI or LVMI/LVEDVI (Table 3). There was no significant relationship between baseline Zva and LVMI nor LVMI/LVEDVI, neither.

### 3.5. The Relationship between ΔZva or ΔMPG and the Change in LV Geometry (ΔLVMI and ΔLVMI/LVEDVI) after TAVR

ΔMPG had a weak correlation with ΔLVMI (ρ = 0.15, *p* = 0.009; Table 3), while ΔZva correlated more strongly with ΔLVMI (ρ = 0.47, *p* < 0.001; *p* for comparison of correlations < 0.001). There was no significant relationship between ΔMPG and ΔLVMI/LVEDVI. In contrast, ΔZva also correlated with ΔLVMI/LVEDVI (ρ = 0.54, *p* < 0.001).

In the LFLG-AS sub-group, a trend indicating a positive correlation between ΔZva and ΔLVMI was observed but was not statistically significant (ρ = 0.38, *p* = 0.083; Table 3). However, there was a significant positive correlation between ΔZva and ΔLVMI/ΔLVEDVI (ρ = 0.51, *p* = 0.015).

### 3.6. Univariate and Multivariate Analysis for the Change in LV Geometries after TAVR (ΔLVMI and ΔLVMI/LVEDVI)

In the univariate and multivariate linear regression analyses to identify the factors associated with ΔLVMI, ΔZva was an independent factor (Table 4). ΔMPG was associated with ΔLVMI in univariate linear regression analysis but not in multivariate linear regression analysis (Table 4).

In univariate and multivariate linear regression analyses to identify the factors associated with ΔLVMI/LVEDVI, ΔZva was an independent factor (Table 5). ΔMPG was not a significant factor associated with ΔLVMI/LVEDVI in neither a univariate nor a multivariate linear regression model (Table 5).

### 3.7. The Comparison of the Measurements between Transthoracic Echocardiography and Computed Tomography

There were good correlations between echocardiographic and CT measurements (LVMI, ρ = 0.87, *p* < 0.001; LVEDVI, ρ = 0.94, *p* < 0.001; SV, ρ = 0.9, *p* < 0.001).

## 4. Discussion

To the best of our knowledge, this study is the first to demonstrate that the improvement in Zva after TAVR is also associated with decreased LVMI/LVEDVI, considered as LV geometrical reverse-remodeling after TAVR. The major findings were as follows: First, the improvement in Zva after TAVR was robustly associated with both decreased LVMI/LVEDVI and LVMI; in contrast, there was a weak association between MPG reduction and LVMI reduction (Table 3). Second, baseline Zva (but not MPG) was strongly associated with higher LVMI/LVEDVI.

### 4.1. The Impact of Zva on LVMI/LVEDVI and LVMI after TAVR

In this study, LVMI/LVEDVI decreased with improvement in Zva after TAVR (Table 2). A recently published report [28] found that increased LVMI/LVEDVI was associated with poor survival in patients with heart failure, including aortic valve stenosis. A significant increase in LVMI after TAVR was observed in patients who died in the first year after TAVR, whereas the survivors had consistent LVMI reduction [10]. Furthermore, a greater decrease in LV mass immediately after TAVR was associated with a lower incidence of re-hospitalization due to heart failure [13]. From the aforementioned collective evidence, decreased LVMI/LVEDVI and LVMI after TAVR are considered as favorable LV remodeling after TAVR. In the present study, the improvement in Zva was strongly associated with both decreased LVMI/LVEDVI and LVMI after TAVR. LFLG-AS has been reported to be associated with a high Zva [23]. In our study population, there were only 22 patients with LFLG-AS. These patients had a high baseline Zva, which decreased after TAVR. ΔZva and ΔLVMI/ΔLVEDVI also showed a positive correlation. A decrease in ΔLVMI/LVEDVI might reflect the improvement of Zva after TAVR. A large number of clinical studies have shown that MPG reduction at 30 days after TAVR is associated with LV mass reduction [13]. In this study, although MPG reduction had a weak relationship with LVMI in the univariate linear regression model, it was not a significant factor in the multilinear regression model (Table 4). Zva should be evaluated alongside MPG in the clinical management of patients after TAVR. When Zva after TAVR does not change or worsens, vascular resistance and systemic hypertension should be carefully evaluated and, when possible, efficiently managed [29].

### 4.2. Sex Difference in LVMI Change after TAVR

The sex difference in LVMI reduction after TAVR has been debated [30,31]. In multivariate linear regression analysis for LVMI change after TAVR, female patients had a more significant decrease in LVMI after TAVR than male patients (Table 4). Chen et al. reported that a more significant LVMI reduction in women than in men was observed after surgical aortic valve replacement [32]. Thus, the sex difference in LVMI change after TAVR might be the same as that seen in surgical aortic valve replacement. Female sex may be a factor influencing LVMI reduction after TAVR.

### 4.3. Limitations

The present study had several limitations. First, echocardiographic follow-up duration might be short for evaluating left ventricular change after TAVR [33]. Second, we could obtain CT measurements in only 21 patients. Although we confirmed good correlations between TTE and CT measurements, TTE measurements may potentially influence the accuracy of evaluation of the LV geometry. Third, neither central aortic pressure nor radial arterial blood pressure had been measured invasively. Fourth, we could not set a control group, which should have included patients who underwent medical treatment or surgical aortic valve replacement. Lastly, the relevance of LV geometrical changes induced by improvement in Zva on the clinical outcome was not examined. However, high Zva (>5 mmHg/mL/m^2^) was reportedly related to poor two-year survival after TAVR [34]. We could not evaluate the relationship between Zva improvement and all adverse structural and functional changes (such as global longitudinal strain) and thus focused on LVMI, and LVMI/LVEDVI, in AS [35]. Therefore, to confirm the impact of Zva after TAVR on clinical outcome and survival, a long-term follow-up study should be conducted in the future.

Our study design was a single-center retrospective observational study. Therefore, a well-designed further study (longer follow-up for clinical events) is warranted to support the conclusions of our study.

## 5. Conclusions

The improvement in Zva after TAVR was more closely related to LV mass and LVMI/LVEDVI reduction than to MPG reduction. Zva should be evaluated before and after TAVR as well as MPG, because the improvement in Zva may induce favorable LV remodeling.

## Figures and Tables

**Figure 1 jcm-09-03143-f001:**
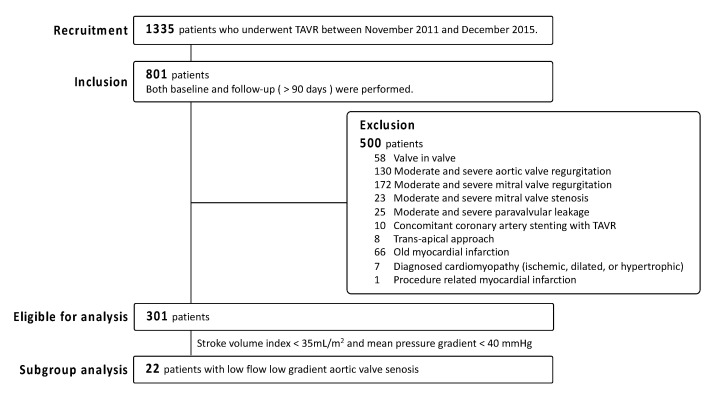
Flowchart of enrollment. TAVR, transcatheter aortic valve replacement.

**Table 1 jcm-09-03143-t001:** Baseline characteristics and procedures.

Characteristics	All Patients (*n* = 301)	LFLG-AS (*n* = 22)
Demographic data		
Age, y	82 ± 8	81 ± 8
Male, *n* (%)	174/301 (58)	10/22 (46)
Body mass index, kg/m^2^	26.9 (23.3, 31.0)	28.9 (25.9, 33.9)
NYHA class ≥ III, *n* (%)	265/301 (88)	19/22 (86)
STS-PROM	4.5 (3.0, 6.1)	4.8 (4.0, 8.8)
Past medical history, *n* (%)		
Diabetes mellitus	109/301 (36)	11/22 (50)
Dyslipidemia	191/301 (64)	11/22 (50)
Hypertension	227/301 (75)	15/22 (68)
Peripheral vascular disease	32/301 (11)	3/22 (14)
Coronary artery disease	118/301 (39)	12/22 (55)
Atrial fibrillation	84/301 (28)	11/22 (50)
Hemodialysis	7/301 (2.3)	22/22 (100)
Obesity (BMI ≥ 30 kg/m^2^)	91/301 (30)	10/22 (46)
Chronic obstructive pulmonary disease	84/301 (28)	8/22 (36)
Pacemaker	34/301 (11)	3/22 (14)
Medication, *n* (%)		
ACE-I/ARB	112/301 (37)	5/22 (23)
Beta-blocker	111/301 (37)	7/22 (32)
Laboratory		
Creatinine, mg/dL	1.1 (0.9, 1.3)	1.4 (0.8, 1.6)
Brain natriuretic peptide, pg/mL	143 (71, 266)	370 (147, 736)
Procedure		
Transaortic approach *, *n* (%)	10/301 (3.3)	1/22 (4.5)
Stent-valve size ≥ 27 mm, *n* (%)	32/301 (11)	2/22 (9.1)
Pacemaker implantation as a complication, *n* (%)	32/301 (11)	4/22 (18)

ACE-I/ARB, angiotensin converting enzyme inhibitor and/or angiotensin receptor blocker; LFLG-AS, low-flow low-gradient aortic valve stenosis; NYHA, New York Heart Association; STS-PROM, Society of Thoracic Surgeons Predicted Risk of Mortality. Appendix A shows the valve system used. Continuous variables with a normal distribution were expressed as mean ± SD, and continuous variables with a skewed distribution were expressed as median (25%, 75%). Categorical variables were expressed as a number (%). * Apart from the transaortic approach, another approach was the femoral approach.

**Table 2 jcm-09-03143-t002:** Valvuloarterial impedance, hemodynamics, and echocardiographic parameters before and after transcatheter aortic valve replacement.

	Overall			LFLG-AS		
	*n* = 301			*n* = 22		
	Baseline	Follow-Up	*p* Value	Baseline	Follow-Up	*p* Value
Zva, mmHg/mL/m^2^	4.91 ± 1.11	3.87 ± 0.81	<0.001	5.61 ± 0.86	4.14 ± 0.98	<0.001
SBP, mmHg	135 ± 22	137 ± 20	0.14	131 ± 20	124 ± 20	0.19
DBP, mmHg	67 ± 13	63 ± 13	<0.001	68 ± 12	62 ±17	0.33
Heart rate, bpm	72 ± 14	71 ± 14	0.29	78 ± 13	75 ± 13	0.5
LVEDVI, mL/m^2^	64 (57, 74)	67 (59, 76)	0.021	62 (52, 70)	67 (53, 75)	0.43
LVESVI, mL/m^2^	26 (19, 34)	27 (21, 35)	0.38	34 (22, 47)	31 (23, 38)	0.28
LV mass index, g/m^2^	99 (83, 118)	86 (69, 103)	<0.001	101 (85, 135)	84 (53, 75)	<0.001
LVMI/LVEDVI, g/Ml	1.51 (1.28, 1.78)	1.27 (1.05, 1.51)	<0.001	1.50 (1.31, 2.12)	1.44 (1.01, 1.52)	<0.001
LV ejection fraction, %	60 (52, 67)	60 (51, 67)	0.52	52 (35, 62)	58 (44, 63)	0.14
Stroke volume index, mL/m^2^	38.2 ± 7.5	39.4 ±7.5	0.011	28.7 ± 4.4	33.5 ± 7.4	0.011
EOAI, cm^2^/m^2^	0.39 ± 0.09	0.88 ± 0.24	<0.001	0.40 ± 0.08	0.83 ± 0.19	<0.001
Patient–prothesis mismatch, *n* (%)						
Insignificant		155/301 (52)			10/22 (46)	
Moderate		98/301 (33)			6/22 (27)	
Severe		48/301 (16)			6/22 (27)	
Peak pressure gradient, mmHg	72 (64, 86)	19 (14, 26)	<0.001	48 (43, 54)	13 (11, 23)	<0.001
Mean pressure gradient, mmHg	44 (40, 52)	10 (8, 14)	<0.001	27 (24, 33)	7 (5, 12)	<0.001
Mild aortic regurgitation, *n* (%)	229 (76)			17/22 (77)		
Mild paravalvular leakage, *n* (%)		178/301 (59)			13/22 (59)	
E wave, cm/sec	88 (71, 110)	96 (76, 122)	<0.001	104 (86, 127)	107 (92, 133)	<0.014
A wave, cm/sec	106 (86, 125)	114 (95, 133)	<0.001	105 (85, 126)	123 (82, 148)	0.68
E/A	0.78 (0.65, 1.06)	0.78 (0.66, 0.97)	0.86	0.78 (0.62, 1.15)	0.86 (0.7, 1.19)	0.24
E’, cm/sec	7.2 (5.7, 8.9)	6.8 (5.4, 8.8)	0.16	7.6 (6.4, 8.8)	7.5 (5.9, 9.9)	0.64

AS, aortic valve stenosis; A wave, peak late mitral inflow velocity; DBP, diastolic blood pressure; EOAI, effective orifice area index; E wave, peak early mitral inflow velocity; E’, early mitral valve velocity; LFLG-AS, low-flow low-gradient aortic valve stenosis; LVEDVI, left ventricular mass index; LVMI/LVEDVI, the ratio of left ventricular mass index to left end-diastolic volume index; LVESVI, left ventricular end-systolic volume index; SBP, systolic blood pressure; Zva, valvuloarterial impedance. *p* values for comparison between parameters at baseline and those at follow-up were calculated by a Wilcoxon signed-rank test.

**Table 3 jcm-09-03143-t003:** The relationship between MPG or Zva and LV geometry (LVMI and LVMI/LVEDVI) at baseline and change after TAVR.

**Overall** **(*n* = 301)**	**Baseline LV geometry (LVMI and LVMI/LVEDVI) vs. MPG or Zva**					
		Baseline MPG		Baseline Zva		*p* for comparison
		ρ	*p*	ρ	*p*	
	Baseline LVMI	0.06	0.28	0.006	0.92	0.49
	Baseline LVMI/LVEDVI	0.05	0.41	0.29	<0.001	0.001
	**Change in LV geometry (LVMI and LVMI/LVEDVI) vs. ΔMPG or ΔZva**					
		ΔMPG		ΔZva		*p* for comparison
		ρ	*p*	ρ	*p*	
	ΔLVMI	0.15	0.009	0.47	<0.001	<0.001
	ΔLVMI/LVEDVI	0.06	0.26	0.54	<0.001	<0.001
**LFLG-AS** **(*n* = 22)**	**Baseline LV geometry (LVMI and LVMI/LVEDVI) vs. MPG or Zva**					
		Baseline MPG		Baseline Zva		*p* for comparison
		ρ	*p*	ρ	*p*	
	Baseline LVMI	0.02	0.92	0.11	0.63	0.8
	Baseline LVMI/LVEDVI	0.04	0.85	0.18	0.42	0.68
	**Change in LV geometry (LVMI and LVMI/LVEDVI) vs. ΔMPG or ΔZva**					
		ΔMPG		ΔZva		*p* for comparison
		ρ	*p*	ρ	*p*	
	ΔLVMI	−0.30	0.17	0.38	0.083	0.034
	ΔLVMI/LVEDVI	−0.15	0.51	0.51	0.015	0.034

LV, left ventricle; LVEDVI, left ventricular end-diastolic volume index; LFLG-AS, low-flow low-gradient aortic valve stenosis; LVMI, left ventricular mass index; MPG, mean pressure gradient; TAVR, transcatheter aortic valve replacement; Zva, valvuloarterial impedance; ρ, Spearman’s rho. Δ denotes the change in value from baseline to follow-up after transcatheter aortic valve replacement.

**Table 4 jcm-09-03143-t004:** Univariate and multivariate linear regression analyses to identify factors associated with ΔLVMI.

	Univariate				Multivariate			
Factor	Β	95% CI		*p* Value	β	95% CI		*p* Value
ΔZva, mmHg/mL/m^2^	9.29	7.11	11.47	<0.001	8.71	5.27	12.15	<0.001
ΔMPG, mmHg	0.26	0.05	0.46	0.016	0.03	−0.16	0.22	0.73
ΔSVI mL/m^2^	−1.1	−1.44	−0.75	<0.001	0.02	−0.48	0.52	0.94
Age, y	−0.29	−0.65	0.07	0.12	−0.24	−0.54	0.06	0.11
Male	2.93	−2.81	8.68	0.32	7.7	2.95	12.44	0.002
Body mass index, kg/m^2^	−0.1	−0.55	0.35	0.67	−0.08	−0.46	0.31	0.7
Diabetes mellitus	1.93	−3.98	7.84	0.52	0.84	−4.07	5.75	0.74
Hypertension	2.26	−4.33	8.86	0.5	−1.71	−7.01	3.58	0.53
ACE-I/ARB	4.75	−1.11	10.6	0.11	2.09	−2.71	6.89	0.39
Beta-blocker	2.86	−3.02	8.74	0.34	1.82	−2.87	6.52	0.45
Paravalvular leakage	1.39	−4.39	7.17	0.64	1.03	−3.67	5.74	0.67
Baseline LVMI, g/m^2^	−0.37	−0.45	−0.29	<0.001	−0.38	−0.45	−0.31	<0.001
Prosthesis–patient mismatch	9.04	1.86	16.22	0.014	5.33	−0.88	11.53	0.094
Systolic blood pressure, mmHg	0.12	−0.03	0.26	0.12	−0.02	−0.16	0.12	0.77
Diastolic blood pressure, mmHg	0.1	−0.12	0.32	0.36	−0.01	−0.2	0.18	0.9
					*N* = 301			
					Adjusted R^2^ = 0.39			
					*p* < 0.001			

ACE-I/ARB, angiotensin converting enzyme inhibitor and/or angiotensin receptor blocker; CI confidence interval; LVMI, left ventricular mass index; MPG, mean pressure gradient; SVI, systolic volume index; Zva, valvuloarterial impedance. Δ denotes the change in value from baseline to follow-up after transcatheter aortic valve replacement.

**Table 5 jcm-09-03143-t005:** Univariate and multivariate linear regression analyses to identify factors associated with ΔLVMI/LVEDVI.

	Univariate				Multivariate			
Factor	Β	95% CI		*p* Value	β	95% CI		*p* Value
ΔZva, mmHg/mL/m^2^	0.19	0.15	0.23	<0.001	0.1	0.04	0.15	<0.001
ΔMPG, mmHg	0	0	0.01	0.15	0	0	0	0.86
ΔSVI mL/m^2^	−0.03	−0.03	−0.02	<0.001	−0.01	−0.02	0	0.049
Age, y	−0.01	−0.01	0	0.035	0	−0.01	0	0.86
Male	−0.01	−0.12	0.09	0.82	0.04	−0.04	0.11	0.35
Body mass index, kg/m^2^	0	−0.01	0.01	0.78	0	0	0.01	0.18
Diabetes mellitus	0.04	−0.07	0.15	0.47	0.03	−0.05	0.1	0.53
Hypertension	0.07	−0.05	0.19	0.27	−0.01	−0.09	0.08	0.87
ACE-I/ARB	0.08	−0.03	0.19	0.14	0.03	−0.05	0.1	0.5
Beta-blocker	0.07	−0.04	0.18	0.19	0.03	−0.05	0.1	0.46
Paravalvular leakage	−0.02	−0.12	0.09	0.75	−0.02	−0.1	0.05	0.51
Baseline LVMI/LVEDVI, g/m^2^	−0.67	−0.76	−0.58	<0.001	−0.61	−0.69	−0.53	<0.001
Prosthesis–patient mismatch	0.19	0.05	0.32	0.006	0.1	0.01	0.2	0.038
Systolic blood pressure, mmHg	0	0	0	0.41	0	0	0	0.11
Diastolic blood pressure, mmHg	0	0	0.01	0.43	0	0	0	0.25
					*N* = 301			
					Adj. R^2^ = 0.56			
					*p* < 0.001			

ACE-I/ARB, angiotensin converting enzyme inhibitor and/or angiotensin receptor blocker; CI confidence interval; LVMI, left ventricular mass index; MPG, mean pressure gradient; SVI, systolic volume index; Zva, valvuloarterial impedance. Δ denotes the change in value from baseline to follow-up after transcatheter aortic valve replacement.

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
