# Peer review of "The Impact of Valvuloarterial Impedance on Left Ventricular Geometrical Change after Transcatheter Aortic Valve Replacement: A Comparison between Valvuloarterial Impedance and Mean Pressure Gradient"

_jcm, 2020, doi:10.3390/jcm9103143_

Round 1
Reviewer 1 Report
All comments were adequately addressed.
Author Response
Reviewer 1
(Comment) All comments were adequately addressed.
(Response) Thank you very much.

Reviewer 2 Report
No further comments.
Author Response
Reviewer 2
(Comment) No further comments.
(Response) Thank you very much.
Reviewer 3 Report
In the present study authors aim at evaluating LV geometrical remodeling after TAVR. They show that the improvement in Zva after TAVR is associated with decreased LVMI/LVEDVI. This is and interesting study and despite of some limitations already stated by the authors, it can represent a good start for further evaluation.
Minor comment: Please indicate the antithrombotic regimen applied in the study population
Author Response
Reviewer 3
(Comment) In the present study authors aim at evaluating LV geometrical remodeling after TAVR. They show that the improvement in Zva after TAVR is associated with decreased LVMI/LVEDVI. This is an interesting study and despite of some limitations already stated by the authors, it can represent a good start for further evaluation.
Minor comment: Please indicate the antithrombotic regimen applied in the study population
(Response) Thank you very much. We add the antithrombotic regimen in the study population, as follows.
(Line 52) Aspirin was initiated at the timing of TAVR and continued at least one year.

This manuscript is a resubmission of an earlier submission. The following is a list of the peer review reports and author responses from that submission.
Round 1
Reviewer 1 Report
The focus of this study is to investigate the relationship between improvement in valvuloarterial impedance and change in LV mass index (LVMI) and the ratio of LVMI to LV end-diastolic volume index. This is an interesting study and I only have a few minor issues.
1) Sex was independently associated with ΔLVMI. The influence of sex should be added to the discussion.
2) Many of the patients were placed on anti-hypertensive drugs at baseline. Please clarify in the texts if the patients remained on the same treatment at follow-up. If there was changes in pharmaceutical treatments, how was this handled in the analysis?
Reviewer 2 Report
The authors investigated the correlation between valvuloarterial impedance and LV reverse remodelling after TAVR. The study is well performed and has a clear message. The study has limitations being retrospective and single center and this should be mentioned in the limitations section.
In the conclusion Zva should be measured not only after TAVR but obviously also before TAVR. I would suggest to add before to the last sentence.
Reviewer 3 Report
Comments are enclosed in the word file
